# Decreased Muscle-to-Fat Mass Ratio Is Associated with Low Muscular Fitness and High Alanine Aminotransferase in Children and Adolescent Boys in Organized Sports Clubs

**DOI:** 10.3390/jcm10112272

**Published:** 2021-05-24

**Authors:** Kai Ushio, Yukio Mikami, Hiromune Obayashi, Hironori Fujishita, Kouki Fukuhara, Tetsuhiko Sakamitsu, Kazuhiko Hirata, Yasunari Ikuta, Hiroaki Kimura, Nobuo Adachi

**Affiliations:** 1Department of Rehabilitation, Hiroshima University Hospital, Hiroshima 734-8551, Japan; ymikami@wakayama-med.ac.jp (Y.M.); luna@hiroshima-u.ac.jp (H.K.); 2Sports Medical Center, Hiroshima University Hospital, Hiroshima 734-8551, Japan; hobay@hiroshima-u.ac.jp (H.O.); h-fujishita@hiroshima-u.ac.jp (H.F.); kouki@hiroshima-u.ac.jp (K.F.); sakamitsut@hiroshima-u.ac.jp (T.S.); hiratakz@hiroshima-u.ac.jp (K.H.); yikuta@hiroshima-u.ac.jp (Y.I.); nadachi@hiroshima-u.ac.jp (N.A.); 3Department of Rehabilitation Medicine, Wakayama Medical University, Wakayama 641-8509, Japan; 4Department of Orthopedic Surgery, Graduate School of Biomedical and Health Sciences, Hiroshima University, Hiroshima 734-8551, Japan

**Keywords:** muscular fitness, exercise, sports clubs, muscle to fat mass ratio, pediatric nonalcoholic fatty liver disease

## Abstract

Decreased muscle-to-fat mass ratio (MFR) is associated with pediatric nonalcoholic fatty liver disease (NAFLD) and may reduce muscular fitness. Regular exercise in sports clubs has not led to reductions in obesity in children and adolescents; they may have decreased MFR. Decreased MFR could cause reduced muscular fitness, which may put them at risk for NAFLD development. We investigated whether MFR is related to muscular fitness and serum alanine aminotransferase (ALT), to determine whether MFR could be used to screen for NAFLD in children and adolescent boys belonging to sports clubs. Altogether, 113 participants (aged 7–17 years) who underwent body composition, laboratory, and muscular fitness measurements during a medical checkup were divided into tertiles according to their MFR. Lower extremity muscular fitness values were significantly decreased in the lowest MFR tertile (*p* < 0.001); conversely, serum ALT levels were significantly increased (*p* < 0.01). Decreased MFR significantly increased the risk of elevated ALT, which requires screening for NAFLD, after adjusting for age, obesity, muscular fitness parameters, and metabolic risk factors (odds ratio = 8.53, 95% confidence interval = 1.60–45.6, *p* = 0.012). Physical fitness and body composition assessments, focusing on MFR, can be useful in improving performance and screening for NAFLD in children and adolescents exercising in sports clubs.

## 1. Introduction

Obesity in children and adolescents is a major problem globally. Pediatric obesity causes decreased muscular fitness [1,2,3,4] and leads to metabolic syndrome and cardiovascular disease in the present [1,4,5] or future [6,7]. Nonalcoholic fatty liver disease (NAFLD) is characterized by an excessive accumulation of fat in the liver in the absence of significant alcohol consumption. NAFLD has a mutual and bidirectional relationship with metabolic syndrome [8] and is the most prevalent chronic liver disease [9]. Recently, NAFLD has become a health problem in children, adolescents, and adults [10]. A recent meta-analysis reported that the global prevalence of pediatric NAFLD is 7.6% in the general population and 34.2% in obese patients, affecting more boys than girls [11]. Pediatric NAFLD follows a more progressive course, such as increased liver fibrosis, compared to adults [12]. In addition, NAFLD is associated with a higher risk of developing diabetes mellitus and cardiovascular disease [13,14]; therefore, early detection and therapeutic intervention are necessary.

Insulin resistance is one of the hallmarks of NAFLD, is key in the pathogenesis of the disease, and is associated with visceral fat accumulation [9]. However, sarcopenia, which is characterized by decreased skeletal muscle mass and physical function, has been reported to affect the development of NAFLD in recent years [15]. In fact, improvement in the muscle-to-fat mass ratio (MFR), rather than improvement in the body mass index (BMI), has led to an improvement in the liver stiffness in adults with NAFLD [16]. An imbalance in the MFR and decreased skeletal muscle mass have also been identified as risk factors for pediatric NAFLD [17,18]. Therefore, it is extremely important to evaluate not only BMI (simple weight-to-height ratio) but also MFR in those with NAFLD.

Exercise along with nutrition is important for the prevention and management of obesity, cardiovascular disease, metabolic syndrome, and NAFLD [19,20,21]. However, performing regular exercise in organized sports clubs does not necessarily lead to a reduction in obesity status (e.g., BMI, body fat, waist circumference) in children and adolescents [22,23,24]. Excessive fat accumulation due to obesity and the relative loss of skeletal muscle mass cause decreased muscular fitness. Therefore, decreased MFR would also be seen in children and adolescents who exercise in organized sports clubs and may be associated with reduced muscular fitness and the development of NAFLD.

Serum alanine aminotransferase (ALT) is recommended as the best screening test for pediatric NAFLD according to the North American Society for Pediatric, Gastroenterology, Hepatology, and Nutrition (NASPGHAN) Clinical Practice Guideline, with substantial limitations (e.g., less than 90% sensitivity, poor specificity) [14]. High levels of serum ALT through childhood and adolescence have also been associated with hepatic fatty degeneration at 24 years of age [25].

No studies have examined the relationship between MFR and muscular fitness, and the relationship between MFR and the development of NAFLD in children and adolescents. If decreased MFR affects muscular fitness and serum levels of ALT, MFR could be used as a training indicator for improving muscular fitness and as a screening indicator for NAFLD. Therefore, we investigated whether MFR is related to muscular fitness and serum ALT levels in children and adolescent boys who belonged to organized sports clubs.

## 2. Materials and Methods

### 2.1. Participants

Participants were 130 children and adolescent boys, aged 7–17 years, who underwent medical checkups at Hiroshima University Hospital in 2018 and 2019 and who belonged to organized sports clubs and exercised regularly (Figure 1). The Hiroshima City Sports Association requested our assistance to conduct medical checkups and had selected all participants. More than 60% of the participants have participated in national competitions. Preliminary questionnaires (Appendix A) and interviews confirmed that all participants exercised at least 60 min a day, at least three times a week in addition to exercise at school, and that there was no past history of cardiovascular disease or metabolic disease, including liver disease. In addition, past and present injuries were evaluated prior to muscular fitness measurements (Figure 1).

This study was approved by the Ethics Committee of Hiroshima University Hospital (file number: E-941). Consent forms were obtained from the guardians and instructors of all participants.

### 2.2. Body Composition Measurements

The participants did not skip breakfast and had fasted for at least 9 h after dinner. They were also not permitted to drink water 30 min prior to body composition measurements, and the measurements were performed after they had urinated. Height (cm) was measured with a Seca 213 portable stadiometer (Seca NIHON, Chiba, Japan) in 1 mm increments, and body weight (kg) was measured with a FitScan FS-101 body composition monitor (TANITA, Tokyo, Japan) in 100-g increments. For the measurements, the participants wore light clothing and no shoes. This was followed by a multifrequency bioelectrical impedance analysis with the InBody S10 water analyzer (InBody Co., Seoul, Korea). The participants lay in a supine position, with the lower limbs apart and the upper limbs not in contact with the trunk. After removing the sebum from the skin in contact with the electrodes using electrolytic tissues (InBody Tissue, InBody Co., Seoul, Korea), the electrodes were placed on the first and third toes and on both ankles. During the measurement, the participants were instructed to lie still and not to speak. BMI, appendicular skeletal muscle mass (ASM), and body fat mass (FM) were calculated based on previously measured data on height and body weight. Furthermore, ASM/weight and FM/weight were calculated by dividing ASM and FM by weight, respectively. The muscle-to-fat mass ratio (MFR) was calculated by dividing ASM by FM. The participants were classified into the following three groups according to their MFR score: <25th percentile was low MFR, 25–75th percentile was medium MFR, and >75th percentile was high MFR. In Japan, individuals with BMI > 25 kg/m^2^ were considered to have obesity, and the proportion of participants with a BMI > 25 kg/m^2^ was examined among the MFR tertiles.

### 2.3. Laboratory Measurements

Before the muscular fitness measurements, venous blood sampling was performed in a fasting state at least 9 h after the last meal. The levels of serum liver enzymes such as ALT, aspartate aminotransferase, and gamma-glutamyltransferase, along with the levels of metabolic risk factors, such as fasting blood sugar (FBS), triglyceride (TG), and high-density lipoprotein cholesterol (HDL-C) were recorded. Criteria for suspicion of NAFLD were defined as ALT > 26 U/L based on the NASPGHAN Clinical Practice Guideline [13].

### 2.4. Muscular Fitness Measurements

The muscular fitness measurements including hand grip for upper limb muscle strength, isokinetic knee extension and flexion force for lower limb muscle strength, and squat jump and counter movement jump for lower limb power were evaluated. A Smedley-style digital grip dynamometer (Grip-D, Takei Scientific Instruments Co., Ltd., Nigata, Japan) was used for the hand grip. The participant performed maximal effort gripping twice in the standing position. The peak value was defined as hand grip. Biodex system 4 (Biodex Medical Systems, New York, NY, USA) was used for the knee extension and flexion force. The participants performed the knee extension and flexion force five times in the sitting position at an angular velocity of 60° per s. The highest peak torque among the five trials was the measured value. An optical measurement system, OptoJump Next (Microgate Co., Bozen, Italy), was used for the squat jump and counter movement jump. The participants were instructed as follows: “From an upright position with your hands on your waist, bend your knees, stand still for a moment, then jump straight up” for the squat jump, and “From an upright position with your hands on your hips, bend your knees and jump straight up” for the counter movement jump. The participants performed these muscular fitness exercises at maximal effort three times. The highest jump heights in the three trials were recorded.

### 2.5. Statistical Analysis

Continuous variables are presented as mean ± standard deviation or median (interquartile range), and percentages are presented as numbers (LRB%). Comparisons of the measurements among the MFR tertiles were performed using the Kruskal–Wallis test. The percentages of obesity (BMI > 25 kg/m^2^) and ALT > 26 U/L were calculated using the chi-square test. In addition, comparisons of ASM, FM, ASM/weight, FM/weight, muscular fitness measurements, and laboratory measurements among the MFR tertiles were conducted using the Steel–Dwass test. To investigate the risk of NAFLD among the MFR tertiles, multiple logistic regression analysis was performed with age, obesity (BMI > 25 kg/m^2^), MF, and metabolic risk factors (FBS, TG, HDL-C) as covariates. Statistical significance was set at *p* < 0.05. The statistical analysis software used was Stata/MP (version 15.1; Stata Corporation, College Station, TX, USA) and JMP (version 15.0; SAS Institute Inc., Cary, NC, USA).

## 3. Results

Body composition and laboratory measurements were performed for all the participants. However, participants with knee or ankle injuries did not undergo muscular fitness measurements in the lower extremities. Eventually, out of 130 participants, 113 were able to perform all measurements and were included in this study.

Characteristics of the study participants are shown in Table 1. There were significant differences in body weight (*p* = 0.0023), BMI (*p* < 0.0001), FM (*p* < 0.0001), ASM/weight (*p* < 0.0001), FM/weight (*p* < 0.0001), knee extension force (*p* = 0.0029), knee flexion force (*p* = 0.0068), squat jump (*p* < 0.0001), counter movement jump (*p* < 0.0001), ALT (*p* = 0.0028), and TG (*p* = 0.048) among the MFR tertiles. There was a significant difference in the percentage of obesity (BMI > 25 kg/m^2^) (*p*< 0001) and ALT > 26 U/L (*p* < 0.0001).

Figure 2 shows the differences in ASM, FM, SMM/weight, and FM/weight among the MFR tertiles. In ASM/weight, there was a significant decrease from low to high MFR values (*p* < 0.001). Conversely, in FM and FM/weight, there was a significant increase from low to high MFR values (*p* < 0.001).

Figure 3 shows the differences in muscular fitness measurements among the MFR tertiles. For knee extension force, the low MFR tertile showed significantly lower values than the medium and high MFR tertiles (*p* < 0.01). Regarding knee flexion force, the low MFR tertile showed significantly lower values than the medium MFR tertile (*p* < 0.01). For squat jump and counter movement jump, the low MFR tertile showed significantly lower values than the medium and high MFR tertiles (*p* < 0.001).

Figure 4 shows the differences in laboratory measurements among the MFR tertiles. Regarding serum ALT levels, the low MFR tertile showed a significantly higher level than the medium (*p* < 0.05) and high MFR (*p* < 0.01) tertiles. No other significant differences were observed.

As shown in Table 2, for ALT > 26 U/L, which may be used to screen for NAFLD, multiple logistic regression analysis revealed that the decrease among the MFR tertiles was significantly higher (*p* < 0.05) in Model 1 [age], Model 2 [age, obesity (BMI > 25 kg/m^2^)], Model 3 [age, obesity (BMI > 25 kg/m^2^), muscular fitness], Model 4 [age, obesity (BMI > 25 kg/m^2^), metabolic risk factors (FBS, TG, HDL-C)], and Model 5 [age, obesity (BMI > 25 kg/m^2^), muscular fitness, metabolic risk factors (FBS, TG, HDL-C)].

## 4. Discussion

This study examined for the first time the relationship between MFR and muscular fitness and the relationship between MFR and serum ALT levels in children and adolescent boys who belonged to organized sports clubs. For those in the lowest MFR tertile, muscle strength and power in the lower extremities decreased, and serum ALT levels were significantly increased. Furthermore, a decrease in the MFR was a possible risk factor for an increase in serum ALT levels, at which screening for NAFLD could be recommended.

There was no difference in ASM among the MFR tertiles. However, FM significantly increased with decreasing MFR. There have been no studies with large population data on ASM, FM, and MFR in Japanese children and adolescents reported in the literature. However, studies involving large data sets of children and adolescents in general in China [26] and Korea [27] have been conducted. The data regarding ASM, FM, and MFR are shown in Table 3. Although the participants in our study were younger, the mean ASM in all MFR tertiles was higher than in those conducted in China and Korea. This suggests that the participants in our study maintained a high level of ASM through daily exercise. On the other hand, FM was higher in the lowest MFR tertile of our study than the Korean average, and the MFR was lower in the lowest MFR tertile in our study than the Korean average. In Japanese adult males with NAFLD and type 2 diabetes mellitus, the mean MFR was 1.43 [28]. This corresponded to the 25th percentile of the lowest MFR tertile in our study. NAFLD cutoff values for children and adolescents with MFR have not been clarified. However, in the lowest MFR tertile, the mean MFR value was lower than the mean value for general Korean boys, and the 25th percentile corresponded to the mean value for NAFLD in Japanese adult males, suggesting that the lowest MFR tertile has a particularly high need for screening for NAFLD.

The lowest MFR tertile showed significantly lower muscular fitness measurement values, other than hand grip, than the medium and high MFR tertiles. Adiposity and muscular fitness are considered to be negatively correlated [1,4]. However, a positive correlation has been noted between adiposity and hand grip, which is explained by the fat-free mass level [29,30]. There was no difference in ASM among the MFR tertiles in our study. This could be the reason why there was no difference in hand grip. In contrast, knee extension force was lower in the obese group than in the healthy group, despite the higher fat-free mass [31]. Jump is a weight-bearing activity that has been negatively correlated with adiposity [4,6]. Therefore, if the ASM levels are similar, a decrease in MFR due to excessive FM is likely to reduce muscle strength and power in the lower extremities. These results suggest that we need to focus on the balance between ASM and FM, as well as adiposity, when assessing muscular fitness in children and adolescent boys who belong to organized sports clubs.

Obesity and overweight are associated with higher rates of NAFLD [10]. In children and adolescents with a BMI above the 85th percentile, screening for NAFLD is recommended by the NASPGHAN Clinical Practice Guideline [14]. However, recently, decreased ASM values and ASM/FM imbalance have been identified as risk factors for NAFLD in children and adolescents with or without obesity [17,18]. NAFLD in non-obese adolescents has been associated with a high rate of sarcopenia [18], defined as an MFR value of ≤1.155 [27]. When this criterion was used, the rate of sarcopenia was low; only 2.7% (3/113) in our study. In fact, there was no difference in ASM values among the MFR tertiles, and these values were higher than the average ASM of general children and adolescents of Asian ethnicity. Furthermore, a decrease in MFR was associated with a higher risk of elevated ALT, which requires screening for NAFLD, after adjusting for age, obesity, muscular fitness, and metabolic risk factors. Therefore, screening for NAFLD by focusing on a decrease in MFR is likely to be important in children and adolescent boys who belonged to organized sports clubs and who exercise regularly, regardless of whether they have obesity (assessed by BMI) or sarcopenia.

Resistance training and aerobic exercise at vigorous or moderate-to-vigorous intensity for 30 min per session, at least three times per week under supervision, are effective in improving hepatic steatosis in children and adolescents with NAFLD, as well as improving muscular strength in a systematic review and meta-analysis [19]. The World Health Organization also emphasizes the importance of vigorous or moderate-to-vigorous-intensity aerobic exercise and muscle- and bone-strengthening exercises to prevent metabolic syndrome and cardiovascular diseases [32]. However, while children and adolescent participation in sports has been linked to higher physical activity levels, it does not necessarily improve obesity [22,23,24]. Vella et al. [24] suggested that leaders and educators should encourage obese children and adolescents to participate in a variety of sports activities. In view of this, it is probable that each sports club has issues such as unbalanced exercise programs and insufficient intensity. Therefore, it is highly possible that we need to assess both body composition and muscular fitness measurement parameters to estimate the bias in the type of exercise, and to prescribe appropriate exercise. We conclude that these efforts can improve muscular fitness and lead to the prevention and management of metabolic diseases and NAFLD. This approach could be applied to children and adolescents in general.

There were some limitations to this study. First, we did not evaluate NAFLD using imaging evaluation or biopsy. The purpose of this study was to screen for NAFLD. In screening, the serum ALT test is recommended by the NASPGHAN Clinical Practice Guideline [14]. However, a non-invasive liver fibrosis test such as transient elastography has been performed in tertiary care settings [33], and recommended screening may change in the future. Second, multifrequency bioelectrical impedance analysis was used to evaluate body composition. Dual energy X-ray absorptiometry is considered the standard for the evaluation of ASM. However, values evaluated with multifrequency bioelectrical impedance analysis have been verified to correlate well with those obtained via dual energy X-ray absorptiometry [34,35]. In addition, multifrequency bioelectrical impedance analysis along with the serum ALT test provides a safe and simplified test, which is suitable for screening. Third, we did not screen for viral infections or alcohol consumption that could affect the liver enzyme levels. However, in Japan, the legal drinking age is >20 years, and underage drinking is strictly monitored. Furthermore, the incidence of hepatitis B and C virus infections is very low in Hiroshima Prefecture, Japan [36]. Fourth, because of the cross-sectional design of this study, we could not determine a causal relationship between muscular fitness and NAFLD and MFR. Therefore, further longitudinal studies with imaging examinations and appropriate exercise interventions are needed in the future. Fifth, our participants were solely children and adolescent boys in Japan with exercise habits. Pediatric NAFLD is more common in boys and Asian races [11]. It is not difficult to conclude that NAFLD is more common in those without exercise habits and those with decreased physical fitness levels. Therefore, our assessments can be applied to children and adolescent boys in general, and even in individuals of different races. However, our finding needs to be investigated in Japanese girls with exercise habits.

## 5. Conclusions

In our study, we clarified that a decrease in MFR was associated with an excessive reduction in muscle strength and power in the lower extremities, and an increase in elevated serum ALT levels, at which screening for NAFLD could be recommended, in children and adolescent boys belonging to organized sports clubs. We conclude that both assessment of physical fitness parameters and body composition, focusing on the balance between ASM and FM, can be useful in improving performance and health management through appropriate exercise instruction. In the future, we would like to expand the scope of our research to include children and adolescents in general and conduct longitudinal interventions based on multiple assessments, including nutritional assessments, to contribute to the health of children and adolescents.

## Figures and Tables

**Figure 1 jcm-10-02272-f001:**
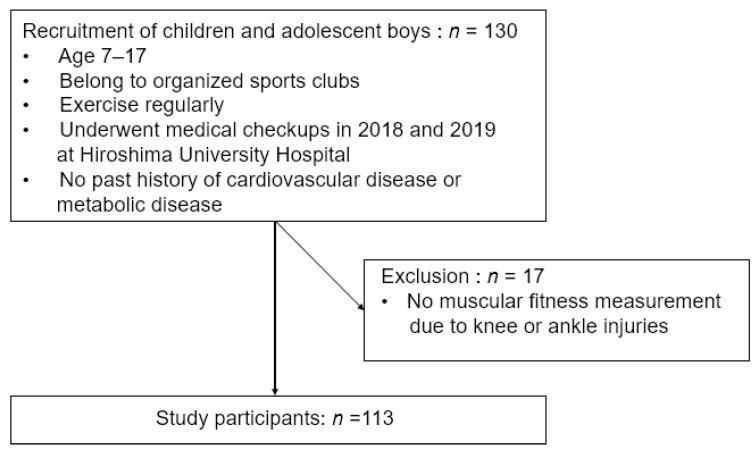
Inclusion and exclusion of participants.

**Figure 2 jcm-10-02272-f002:**
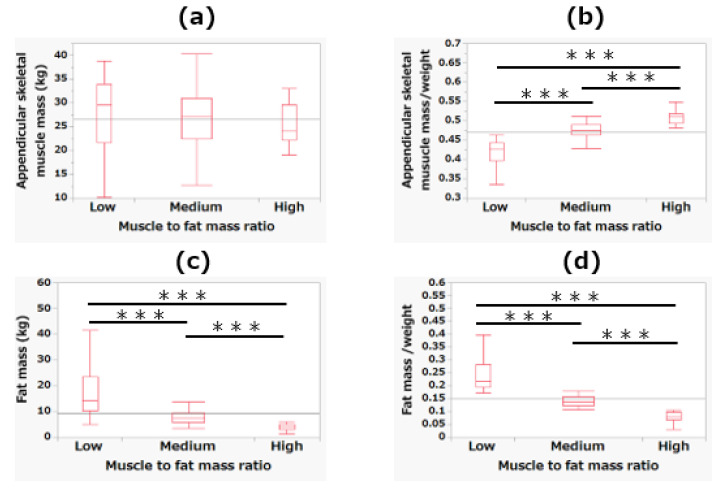
The differences among the tertiles of skeletal muscle-to-fat mass ratio in skeletal muscle mass (**a**), skeletal muscle mass/weight (**b**), fat mass (**c**), and fat mass/weight (**d**). *** *p* < 0.001.

**Figure 3 jcm-10-02272-f003:**
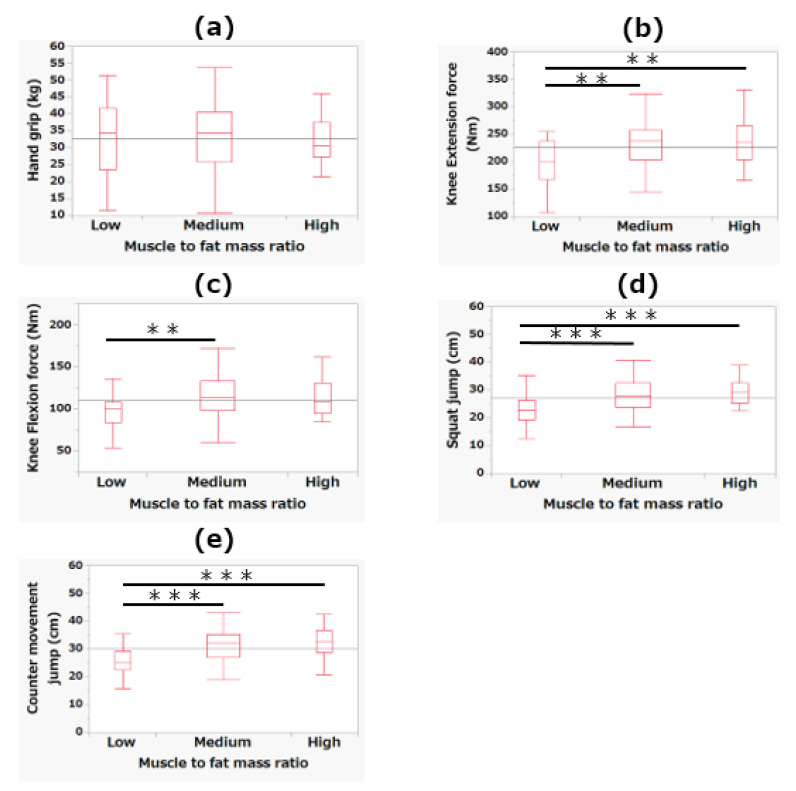
The differences among the tertiles of muscle-to-fat mass ratio in hand grip (**a**), knee extension force (**b**), knee flexion force (**c**), squat jump (**d**), and counter movement jump (**e**). ** *p* < 0.01; *** *p* < 0.001.

**Figure 4 jcm-10-02272-f004:**
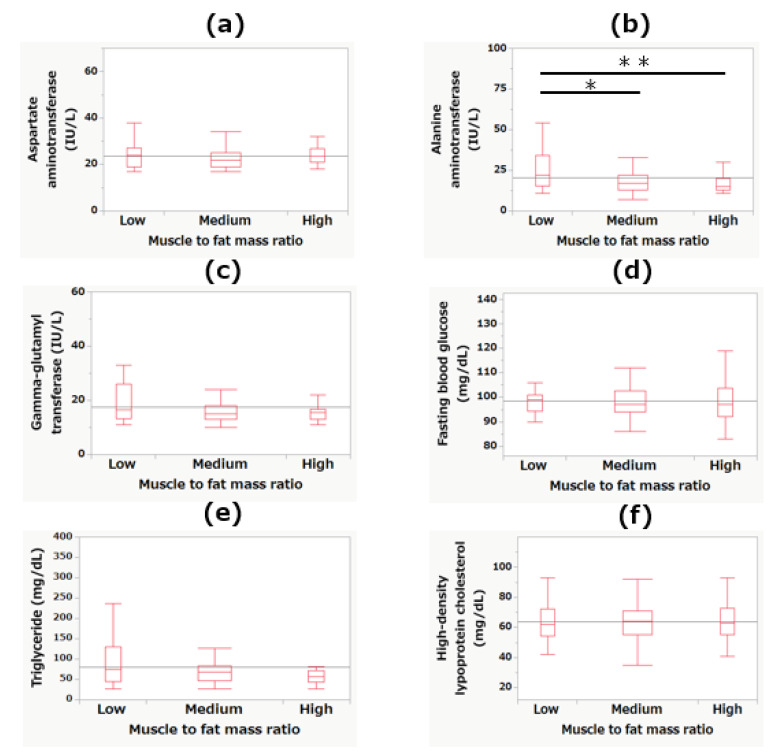
The differences among the tertiles of muscle-to-fat mass ratio in levels of aspartate aminotransferase (**a**), alanine aminotransferase (**b**), gamma-glutamyl transferase (**c**), fasting blood glucose (**d**), triglyceride (**e**), and high-density lipoprotein cholesterol (**f**). * *p* < 0.05; ** *p* < 0.01.

**Table 1 jcm-10-02272-t001:** Characteristics of study participants among the muscle-to-fat mass ratio tertiles.

	ALL (*n* = 113)	Low MFR (*n* = 28)	Medium MFR (*n* = 57)	High MFR (*n* = 28)	*p*-Value
Age, year	14.0 (13.0–15.0)	14.0 (12.3–15.8)	14.0 (13.0–15.0)	14.0 (13.0–15.0)	0.77
Height, cm	165.7 (158.4–171.8)	166.9 (157.6–170.3)	166.5 (157.9–173.2)	165.0 (158.9–170.0)	0.9
Weight, kg	55.7 (44.9–65.0)	68.7 (51.7–80.8)	56.8 (48.4–63.0)	48.1 (44.2–57.8)	0.0023
Body mass index, kg/m^2^	19.7 (17.9–22.4)	24.9 (19.5–28.0)	20.2 (18.0–22.0)	18.1 (16.6–19.3)	<0.0001
Skeletal muscle mass, kg	26.6 ± 7.0	27.3 ± 8.2	26.7 ± 7.4	25.6 ± 4.4	0.44
Fat mass, kg	7.2 (4.9–11.0)	14.3 (10.2–23.4)	7.4 (5.8–9.4)	4.0 (3.1–5.1)	<0.0001
Skeletal muscle mass/weight	0.48 (0.45–0.49)	0.43 (0.40–0.44)	0.48 (0.46–0.49)	0.51 (0.49–0.10)	<0.0001
Fat mass/weight	0.14 (0.11–0.18)	0.22 (0.20–0.28)	0.14 (0.12–0.16)	0.08 (0.07–7.8)	<0.0001
Skeletal muscle to fat mass ratio	3.39 (2.57–4.64)	1.94 (1.43–2.28)	3.39 (2.87–4.07)	6.38 (5.11–7.42)	<0.0001
Hand grip, kg	32.8 ± 9.5	33.4 ± 10.8	32.8 ± 10.0	32.0 ± 7.1	0.68
Knee extension force, Nm/kg	2.26 ± 0.45	2.00 ± 0.38	2.32 ± 0.43	2.40 ± 0.47	0.0029
Knee flexion force, Nm/kg	1.07 (0.94–1.26)	1.00 (0.84–1.08)	1.13 (0.99–1.33)	1.10 (0.95–1.31)	0.0068
Squat jump, cm	27.0 ± 6.1	22.5 ± 5.1	28.2 ± 5.7	29.0 ± 5.4	<0.0001
Counter movement jump, cm	30.0 ± 6.0	25.6 ± 4.9	31.2 ± 5.7	32.3 ± 5.6	<0.0001
Aspartate aminotransferase, IU/L	23.0 (19.5–25.5)	24.0 (19.0–27.0)	22.0 (19.0–25.0)	23.5 (21.0–26.8)	0.34
Alanine aminotransferase, IU/L	17.0 (14.0–23.0)	22.0 (15.3–34.0)	17.0 (13.0–22.0)	15.0 (13.0–20.0)	0.0028
Gamma-glutamyl transferase, IU/L	15.0 (13.0–19.5)	16.5 (15.3–34.0)	15.0 (13.3–26.0)	15.5 (13.0–16.8)	0.16
Fasting blood glucose, mg/dL	98.0 (94.0–102.5)	99.0 (94.3–101.0)	97.0 (94.0–102.5)	97.0 (92.3–103.8)	0.85
Triglycerides, mg/dL	68.0 (45.0–82.5)	74.5 (45.3–130.0)	68.0 (46.0–82.5)	57.0 (43.8–71.3)	0.048
High-density lipoprotein cholesterol, mg/dL	63.9 ± 12.2	63.7 ± 13.1	63.8 ±12.1	63.9 ± 12.2	0.96
Obesity (body mass index > 25 kg/m^2^)	18 (15.9%)	13 (46.4%)	5 (8.8%)	0 (0.0%)	<0.0001
Alanine aminotransferase (>26 IU/L)	16 (15.0%)	11 (39.3%)	4 (7.0%)	1 (3.6%)	<0.0001

Results are presented as mean ± standard deviation, median (interquartile range), or number (LRB%). Low MFR: <25th percentile; medium MFR: 25th–75th percentile; high MFR: >75th percentile. Abbreviations: MFR, muscle-to-fat mass ratio; *n*, number.

**Table 2 jcm-10-02272-t002:** Adjusted odds ratio (95% confidence interval) of decreased muscle-to-fat mass ratio tertiles for serum alanine aminotransferase >26 U/L (screening indicator for NAFLD).

	Decreased MFR Tertiles	*p*-Value
Model 1	7.10 (2.37–21.3)	<0.001
Model 2	5.69 (1.67–19.3)	0.005
Model 3	4.62 (1.19–18.0)	0.027
Model 4	9.32 (1.95–44.7)	0.005
Model 5	8.53 (1.60–45.6)	0.012

Model 1: adjusted for age; Model 2: adjusted for age and obesity (BMI > 25 kg/m^2^); Model 3: adjusted for age, obesity (BMI > 25 kg/m^2^), hand grip, knee extension, knee flexion, squat jump, and counter movement jump; Model 4: adjusted for age, obesity (BMI > 25 kg/m^2^), fasting blood glucose, triglycerides, and high-density lipoprotein cholesterol; Model 5: adjusted for age, obesity (BMI > 25 kg/m^2^), hand grip, knee extension, knee flexion, squat jump, counter movement jump, fasting blood glucose, triglycerides, and high-density lipoprotein cholesterol. Abbreviations, MFR: muscle-to-fat mass ratio.

**Table 3 jcm-10-02272-t003:** Appendicular skeletal muscle mass, fat mass, and muscle-to-fat mass ratio in general boys in China [26] and Korea [27].

	China ^†^	Korea ^§^	ALL Participants (Present Study)	Lowest MFR Tertile (Present Study)
Skeletal muscle mass, kg	22.1 ± 3.6	22.4 ± 0.4	26.6 ± 7.0	27.3 ± 8.2
Fat mass, kg	16.3 ± 8.1	13.9 ± 0.9	7.2 (4.9–11.0)	14.3 (10.2–23.4)
Muscle-to-fat mass ratio	–	2.1 ± 0.1	3.39 (2.57–4.64)	1.94 (1.43–2.28)

^†^ The maximum values at 16 years are presented in boys aged 3 to 17 years. ^§^ The maximum skeletal muscle mass at 17 years, fat mass at age 16 years, and skeletal muscle to fat mass ratio value at age 15 years are presented in boys aged 10 to 17 years. Data are presented as mean ± standard deviation. Abbreviations, MFR: muscle-t-fat mass ratio.

## Data Availability

Data are available upon reasonable request by contacting the corresponding author.

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
