# Peer review of "Decreased Muscle-to-Fat Mass Ratio Is Associated with Low Muscular Fitness and High Alanine Aminotransferase in Children and Adolescent Boys in Organized Sports Clubs"

_jcm, 2021, doi:10.3390/jcm10112272_

Round 1
Reviewer 1 Report
GENERAL COMMENT
Sarcopenia, a definite risk factor of NAFLD development and progression in adults and elderly (Hepatology. 2017 Dec;66(6):2055-2065. Can J Gastroenterol Hepatol. 2020 Oct 31;2020:8859719., has comparatively less characterized in pediatric NAFLD. With this backset, submission JCM-12066004 follows an innovative purpose and is a timely article. This study als has major limitations, the most fundamental of which is, in my view, using a surrogate index (ALT) which is non-specific and poorly sensitive rather than then the less invasive and more accurate imaging techniques (Dig Liver Dis. 2017 May;49(5):471-483. ). Additionally, this article's legibility must be improved and some miscoceptions reworked such as detailed below.
SPECIFIC COMMENT
MAJOR
The strongest limitation of this study is that it recruits a relatively limited number of boys. Given that evaluation of sex differences is a major priority of NAFLD research, this limitation must be accurately discussed.
This study is difficult to follow owing to non-standard abbreviations. For example I see no reason to say “MF” rather than “muscular fitness” throughout the manuscript. Avoiding this abbreviation would facilitate reading. These authors alos use “SFR” for skeletal muscle mass to fat mass ratio. However, this is a non-standard abbreviation which hs commonly used to designate SFR - stone-free rate (Int J Clin Pract. 2021 Apr 16:e14216. Urol Int. 2021 Apr 15:1-8. doi: 10.1159/000506652. Transl Androl Urol. 2021 Mar;10(3):1179-1191. Res Rep Urol. 2021 Mar 23;13:147-154. J Endourol. 2021 Mar 26. doi: 10.1089/end.2020.1128. World J Urol. 2021 Mar 18. doi: 10.1007/s00345-021-03656-y. Online ahead of print. PMID: 33738574) SFR - Sophorae Flavescentis Radix (J Ethnopharmacol. 2021 Mar 26;274:114042. Phytother Res. 2021 Mar 9. doi: 10.1002/ptr.7063. Online ahead of print. PMID: 33751724 ); SFR - spontaneous firing rate - (Otol Neurotol. 2021 Apr 14. doi: 10.1097/MAO.0000000000003183). SFR - salivary flow rate - (BMC Geriatr. 2021 Apr 14;21(1):245.); SFR - Societe Francaise de Rhumatologie (French Rheumatology Society -) - (Joint Bone Spine. 2021 Apr 1:105181. doi: 10.1016/j.jbspin.2021.105181) an SFR - Spotted Fever Rickettsioses Front Public Health. 2021 Mar 12;9:577789. doi: 10.3389/fpubh.2021.577789). Therefore, “ SFR” for skeletal muscle mass to fat mass ratio should either be abandoned or reworked “muscle-to-fat ratio”. Similarly, FFM and KEF should be avoided.
Line 40 “ NAFLD is a phenotype of metabolic syndrome that affects the liver ”. This statement is amenable to criticism in as much as it fails to correctly identify the mutual and bi-directional association of NAFLD with Metabolic Syndrome (for full list of references see Int J Mol Sci. 2020 Aug 16;21(16):5888.).
Line 60 “regular exercise in organized sports clubs does not necessarily lead to a reduction in obesity”. I guess this also depends on how obesity is defined. For example, it is widely acknowledged that BMI performs poorly in identifying obesity in athletes owing to their well developed muscle mass. At any rate, it should be recognized that physical activity is a definite determinant of healthy metabolic state in the obese (Ann Hepatol. 2020 Jul-Aug;19(4):359-366.).
Lines 65-67 although it is important to highlight the novelty of the study “no studies have examined the relationship between 65 MF and SFR, and the relationship between the risk of NAFLD and SFR in children and 66 adolescents in general, as well as in children and adolescents who belong to organized 67 sports clubs.” I recommend to better illustrate the RATIONALE of the study: what (based on NAFLD physiopathology) did these authors expect to find and why ?
Lines 70-71 “High serum ALT concentrations in children and adolescents have also been associated with hepatic fatty degeneration in adults” Sentence unclear. I understand that ALT is important in pediatric NAFLD. As for adults, it should be remembered that over-reliance on transaminases is an acknowledged mistake (J Hepatol. 2013 Oct;59(4):859-71. ).
Line 74; 199-202 - The contention that ALT concentration may serve as a screening tool for NAFLD in children and adolescent boys who belong to organized sports clubs is weak unless supported by bibliographic citations.
Line 273 “In fact, the serum ALT test is 273 an inexpensive blood test that is available in most healthcare centers compared to imaging 274 methods” This is insufficient reason and things may change in a short time (Liver Int. 2021 Apr 24. doi: 10.1111/liv.14908. Epub ahead of print. PMID: 33894100.Curr Pharm Des. 2020;26(32):3915-3927).
MINOR
Throughout the manuscript “non-alcoholic” should best be reworded to “nonalcoholic”. Although both spellings are commonly used, the latter is consistent with pioneering papers that have coined NAFLD and NASH (reviewed in Int J Mol Sci. 2020 Aug 16;21(16):5888.).
Line 13 “with some restrictions” please, declare which these restrictions are.
Line 49 “The main pathogenesis of NAFLD is insulin resistance” Please, reword in standard English.
Lines 50-51 “sarcopenia, which is characterized by decreased skeletal muscle mass and physical function, has affected the development of NAFLD in recent years” Of course, sarcopenia has not affected the development of NAFLD in recent years but, rather, has been acknowledged to do so in recent years.
Lines 58-59 “improvement and prevention of obesity, cardiovascular disease, and metabolic syndrome, including NAFLD” Please, rework as follows “prevention and management of obesity, cardiovascular disease, metabolic syndrome and NAFLD”
Reviewer 2 Report
The study of Ushio et al. links the reduction of skeletal muscle compared to total fat mass in children and adolescent boys, aged 7–17 years, who attend sports clubs and exercised regularly. However, I have my major concern about the study is the assumption that the authors made regarding NAFLD diagnosis. Furthermore, due to high variability in the measures, alanine aminotransferase is not adequate parameter for defining NAFLD, especially in children and adolescents who diet is often not equilibrate and/or regular. I would shift the association found by the study with obesity rather than NAFLD.
Minor concerns:
- For ease of understanding, avoid the excessive use of acronyms
- Lack of CONSORT chart that indicates number of patients, inclusion and exclusion criteria, etc.
- Line 91: better clarification of the “9-hours fasting”. Before or after breakfast?
- Table 1. Hand grip: higher results in low SFR participants?
- Indicate the n in the graphics
- Improve quality of graphics
- Table 2. Improve clarification and description
Round 2
Reviewer 1 Report
Authors tried to address my points but failed in some. Careful editing of English is recommended.
- Line 40 “NAFLD is a phenotype of metabolic syndrome that affects the liver”. This statement is amenable to criticism in as much as it fails to correctly identify the mutual and bi-directional association of NAFLD with Metabolic Syndrome (for full list of references see Int J Mol Sci. 2020 Aug 16;21(16):5888.). Response: We changed this sentence to “NAFLD has a mutual and bidirectional relationship with metabolic syndrome”. (p.1, Line 40-41) ---> add a reference
- Lines 70-71 “High serum ALT concentrations in children and adolescents have also been associated with hepatic fatty degeneration in adults” Sentence unclear. I understand that ALT is important in pediatric NAFLD. As for adults, it should be remembered that over-reliance on transaminases is an acknowledged mistake (J Hepatol. 2013 Oct;59(4):859-71.). Response: No specific changes were made in this sentence, as we think that the original text was clear. I want you to understand that there is no mention of transaminases levels in adults. --> The sentence remains obscure and incomprehensible.
- Line 273 “In fact, the serum ALT test is 273 an inexpensive blood test that is available in most healthcare centers compared to imaging 274 methods” This is insufficient reason and things may change in a short time (Liver Int. 2021 Apr 24. doi: 10.1111/liv.14908. Epub ahead of print. PMID: 33894100.Curr Pharm Des. 2020;26(32):3915-3927). Response: We have deleted this sentence. --> things may change in a short time and this must be said.
- Line 49 “The main pathogenesis of NAFLD is insulin resistance” Please, reword in standard English. Response: We changed the term “pathogenesis” to “etiology” (p.2, Line 49). --> Again this is not standard English.
Reviewer 2 Report
The paper improved its quality. However, the authors still assume that high BMI, decreased ASM value and ASF/FM imbalance means that the subjects are suffering from NAFLD.
